# Histology-specific long-term oncologic outcomes in patients with epithelial ovarian cancer who underwent complete tumor resection: The implication of occult seeds after initial surgery

**Kazumasa Mogi**[1,2☯], **Masato Yoshihara**[1☯*], **Ryo Emoto**[3], **Emiri Miyamoto**[1],
**Hiroki Fujimoto**[1,4], **Kaname Uno**[1,5], **Sho Tano**[1], **Shohei Iyoshi**[1], **Kazuhisa Kitami**[6],
**Nobuhisa Yoshikawa**[1], **Shigeyuki Matsui**[3], **Hiroaki Kajiyama**[1]

1 Department of Obstetrics and Gynecology, Nagoya University Graduate School of Medicine, Nagoya, Aichi, Japan, 2 Department of Medical Genomics Center, Nagoya University Hospital, Nagoya, Aichi, Japan, 3 Department of Biostatistics, Nagoya University Graduate School of Medicine, Nagoya, Aichi, Japan, 4 Robinson Research Institute, Discipline of Obstetrics and Gynaecology, Adelaide Medical School, University of Adelaide, Adelaide, South Australia, Australia, 5 Department of Laboratory Medicine, Division of Clinical Genetics, Lund University Graduate School of Medicine, Lund, Sweden, 6 Department of Gynecologic Oncology, Aichi Cancer Center Hospital, Nagoya, Aichi, Japan

☯ These authors contributed equally to this work.
* myosihara1209@med.nagoya-u.ac.jp

## Abstract

### Objective

Assessing the histology-specific prognosis of epithelial ovarian cancer (OvCa) is clinically challenging, especially in a patient population with a favorable prognosis. This study investigated the histology-specific long-term oncologic outcomes in OvCa patients who underwent complete tumor resection using a large-scale patient cohort form multiple institutions under a central pathological review system.

### Methods

A regional multi-institutional study was conducted from 1986 to 2019. Of the 4,898 patients with ovarian tumors enrolled, 1,175 patients who underwent complete tumor resection were classified into three classes based on clinically important prognostic factors: stage, cytology, ascites volume. For each class category, the effect of histology types on recurrence-free survival, the site of recurrence, and post-recurrence survival was evaluated.

### Results

Recurrence-free survival varied significantly across different histologies (P < 0.001). The risk of recurrence was higher in serous carcinoma compare to other histologies (P < 0.001). The site of tumor recurrence varied by the histology type. Multinominal logistic regression analysis revealed that mucinous histology had a significantly higher likelihood of developing

**Data Availability Statement:** The data that support the findings of this study are available from Nagoya University, Aichi, Japan. But restrictions apply to the availability of these data, which were used under license for the current study, and so are not publicly available. The data will be shared on reasonable request to the Tokai Ovarian Tumor Study Group, Tsuruma-cho 65, Showa-ku, Nagoya University Graduate School of Medicine, Nagoya 466-8550, Aichi, Japan, Phone: +81-52-744-2261; Fax: +81-52-744-2268.

**Funding:** The authors received no specific funding for this work.

**Competing interests:** The authors have declared that no competing interests exist.

recurrent tumors at distant sites from the peritoneum compared to other histologies (P = 0.002). Conversely, serous histology was associated with better post-recurrence survival (Log-rank P < 0.001).

## Conclusions

Long-term oncologic outcomes significantly differ by histology type in OvCa patients who have undergone complete tumor resection at the initial surgery. A careful evaluation of the clinical background is necessary for these patients, and further clinical research into individualized treatment approaches is essential.

## Introduction

Epithelial ovarian cancer (OvCa) is one of the most destructive gynecologic malignancies with more than 300,000 newly diagnosed cases and 200,000 reported deaths worldwide [1]. It is also known as a "silent killer" because most women with OvCa remain asymptomatic until the disease progresses to advanced stages [2]. Consequently, the extent of cytoreductive surgery is one of the major prognostic factors for these patients [3]. Previous studies demonstrated that complete surgical resection without any macroscopic residual tumor improved the prognosis in advanced OvCa [4]. Therefore, gynecologic oncologists have been striving to achieve complete cytoreductive surgery to reduce the risk of tumor recurrence.

Nevertheless, even patients with early-stage disease macroscopically confined to the resected ovary occasionally develop recurrent tumors [5]. This is thought to be due to stem from invisible occult tumor metastasis throughout the body, including in the peritoneal cavity, lymph nodes, and distant parenchymal organs [6]. Recurrence essentially arises from "seeds" of invisible cancer cells that are not successfully removed by intensive treatment or the innate tumor elimination system and are difficult to detect or recognize at the end of a series of initial treatments. Therefore, it is clinically important to clarify the extent to which microscopic occult tumor metastasis influences the oncologic outcomes of patients with OvCa, even after successful complete resection.

OvCa is classically categorized into four major histologies: serous, clear-cell, mucinous, and endometrioid carcinoma [7]. Most of these histologies are considered to originate not from the ovarian epithelium but from other tissues, such as the fallopian tubes and endometrium. Consequently, the OvCa cells of each histology exhibit different genetic, biological, and morphological features [8]. Despite these differences, a universal approach diagnosis, treatment, and follow-up is currently practiced [9], although the validity of this approach, considering the biological characteristics of each histology remains underexplored [10]. This may be partly because serous carcinomas account for the majority of epithelial ovarian cancers, while endometrioid, clear-cell, and mucinous types are less common, making it difficult to conduct comparative studies across different histologies; however, the non-serous histologies are not rare enough to be negligible. Since the incidences of clear-cell and mucinous histologies are higher in Japan than in the United States [2,11], these histologies cannot be overlooked. A more detailed understanding of the clinical characteristics of each histology and the risk of recurrence, mortality, and the frequent site of metastasis is important, especially for patients with the prospect of a long-term prognosis.

Therefore, this study aimed to elucidate the histology-specific long-term oncologic outcomes in OvCa patients without macroscopic residual tumors using a large-scale patient

cohort accumulated in multiple institutions under a central pathological review system. We investigated common prognostic factors in this cohort and assessed histology-specific clinical features related to recurrence-free and post-recurrence survival. Additionally, we examined differences in the site and frequency of histology-specific recurrence, which appeared to be important from the perspective of long-term management.

## Materials and methods

### Study participants

Patients with malignant ovarian tumors were registered between January 1986 and September 2019 using the data of the Tokai Ovarian Tumor Study Group (TOTSG), consisting of Nagoya University Hospital and affiliated institutions. The present study was approved by the Ethics Committee of Nagoya University (approval number 357) and conducted in accordance with the principles of the Declaration of Helsinki. The consent was not obtained because this study is a retrospective data analysis with anonymous information. The authors accessed the registered data from June 22, 2021 for research purposes, had not access to information that could identify individual participants during or after data collection. 4,898 patients with ovarian tumor were identified in this registry system. Those patients had primary surgery between November 1978 and August 2018. Among them, eligible cases included patients who (1) were diagnosed with primary OvCa of the four major histology types (serous, clear-cell, mucinous, and endometrioid carcinoma) based on a central pathological review; (2) received initial surgery and periodic follow-ups at the institutions; and (3) underwent complete tumor resection without macroscopic residual tumors (n = 2098). We excluded patients with distant metastasis (M1) at the initial diagnosis or insufficient information on first-line chemotherapy and the date of recurrence or death. Eligible cases eventually totaled 1,175 (S1 Fig). All histopathological slides were reviewed by an expert pathologist according to the criteria of the World Health Organization classification [7] with no knowledge of patients' clinical data. Sufficient data were available on survival outcomes and clinical staging was performed by the system of the International Federation of Gynecology and Obstetrics [12].

### Surgery, chemotherapy, and follow-up

All patients underwent primary laparotomy to assess the abdominal contents. The procedure principally consisted of hysterectomy and bilateral salpingo-oophorectomy with a full perito-neal evaluation with aspiration or wash cytology, biopsy, and/or omentectomy, staging lym-phadenectomy, and peritoneal exploration. Some patients underwent incomplete surgery, including uterine preservation and the omission of staging lymphadenectomy for clinical reasons, including advanced disease, fertility-sparing, and old age. Details of adjuvant chemotherapy in each time period were described in our previous study [13]. All patients were followed up at each institution every 1–3 months during the first and second years, every 3–6 months during the third to fifth years, and annually until ten years. Follow-up procedures including a gynecological examination, CA125 evaluation, ultrasonographic scan, and periodic radiologic imaging using a computed tomographic scan, magnetic resonance imaging, and/or positron emission tomography. Recurrence was diagnosed based on radiological and/or clinical findings according to the Gynecologic Cancer InterGroup criteria [14]. Recurrence-free survival was defined as the time interval between the date of the initial surgery to that of recurrence, cancer-specific death, or the last follow-up visit. Post-recurrence survival was defined as the time interval between the date of first recurrence to that of cancer-specific death or the last follow-up visit.

## Statistical analysis

Firstly, Cox regression multivariate analysis was applied for selecting the common prognostic factors for recurrence-free survival in this study cohort. According to the prognostic categories, hazard ratio and survival trend for recurrence-free survival were assessed with univariate Cox regression analysis and Kaplan-Meier method with Log-rank test (Fig 1). Secondly, difference of recurrence-free survival trend in all patients and recurrence-free interval in those with recurrent tumor were analyzed by Kaplan-Meier method with Log-rank test and Kruskal-Wallis test with stratification of histologiy types and the classes (Fig 2). Thirdly, the site of recurrence regarding the peritoneum, lymph node, and/or distant organs was compared among the histology types. Additionally, Multinominal logistic regression analyses were used to assess the odds ratio (OR) for tumor recurrence within 10 years at lymph node or distant organs referred to at the peritoneum (Fig 3). Finally, differences of recurrence-after survival trend in patients with tumor recurrence were analyzed by Kaplan-Meier method with Log-rank test with stratification of histology types and the classes. Also, the hazard ratio of recurrence-after survival regarding recurrence within or after 6 months were investigated with multivariate Cox regression analysis adjusted with the classes and stratified by histology types (Fig 4). Significance was set as two-sided with a P value <0.05. All statistical analyses were conducted using IBM SPSS Statistics, Version 28.0 (IBM Corp., Armonk, NY, USA) and GraphPad Prism 10 (V10.1.2) software.

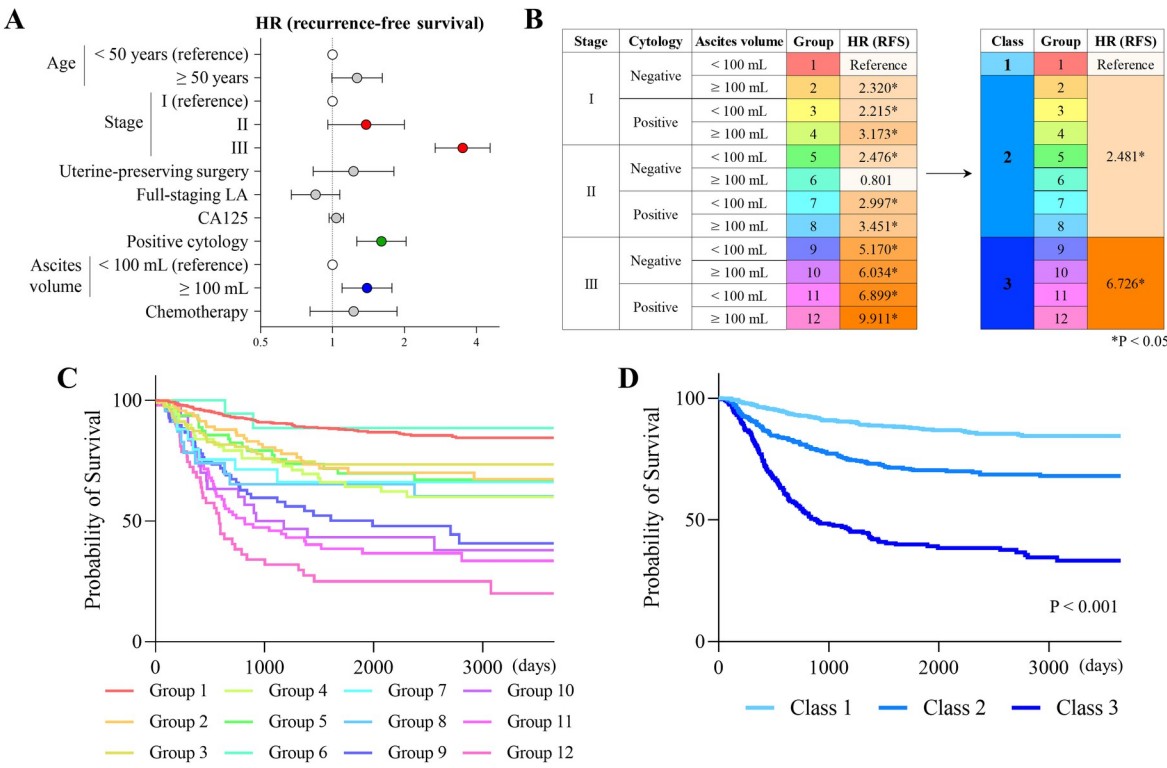

**Fig 1. Histology-independent prognostic factors for recurrence-free survival.** Hazard ratios for recurrence-free survival assessed by a multivariate Cox regression analysis without histological factors (A). Groups categorized by stage, ascites cytology, and ascites volume, and their hazard ratios for recurrence-free survival are shown. According to the levels of hazard ratios, 3 classes and their hazard ratios are also shown (B). Kaplan-Meier curves of recurrence-free survival stratified by 12 groups (C) and 3 classes (D) are shown.

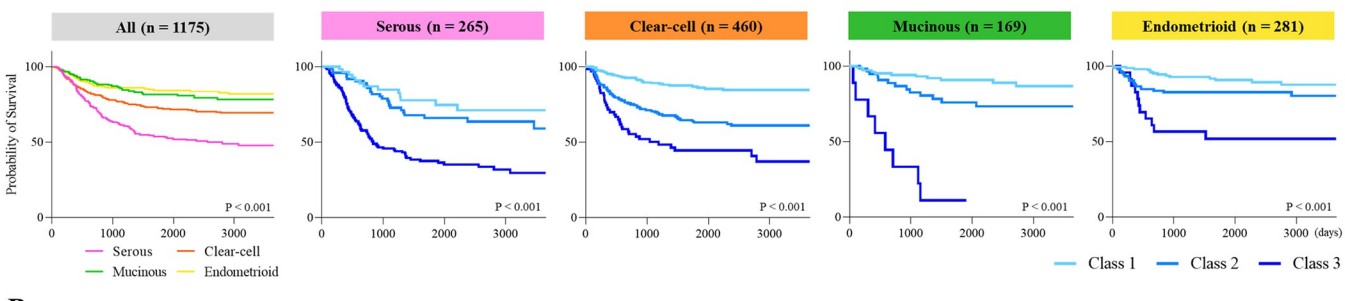

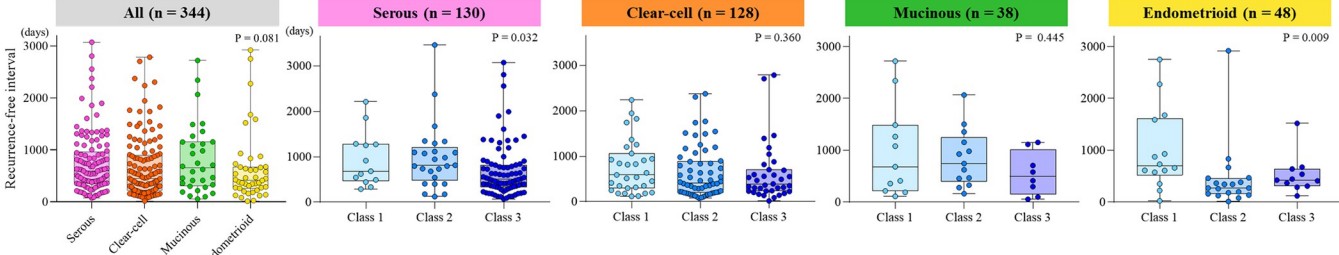

**Fig 2. Effects of histology types on recurrence-free survival.** Kaplan-Meier curves of recurrence-free survival stratified by histology types and the 3 classes (A). P-values were estimated by the Log-rank test. Recurrence-free intervals for patients who developed recurrent tumors were among the 3 classes with the stratification of the histology types (B). P-values were estimated by the Kruskal-Wallis test.

## Results

### Baseline characteristics of patients

Among the study cohort, we included 1,175 patients (Table 1). Histology types were distributed as serous: 265 patients (22.6%), clear-cell: 461 (39.2%), mucinous: 168 (14.3%), and endometrioid: 281 (23.9%). The mean age of patients was 54.2 years. FIGO stages were categorized as stage I: 818 patients (69.6%), stage II: 129 (11.0%), and stage III: 228 (19.4%). Uterine-preserving surgery was performed on 12% of patients (n = 141), while 54.1% (n = 636) underwent complete-staging retroperitoneal lymphadenectomy. Ascites cytology was positive in 30.6% of patients (n = 360) and less ascites (< 100 ml) was confirmed in 76.2% (n = 895). Chemotherapy was performed on 82.3% of patients (n = 967).

### Histology-independent prognostic factors for recurrence-free survival

We explored histology-independent clinical factors associated with recurrence-free survival to classify the study population into simple prognostic categories. The median follow-up was 57.0 months. During the follow-up period, 324 patients (27.6%) developed tumor recurrence and 202 (17.2%) died of the disease. Multivariate analysis revealed that stage, the result of cytology, and ascites volume (< 100 ml, ≥ 100ml) significantly affected recurrence-free survival (Fig 1A and S1 Table). We then categorized the population into 12 groups according to a combination of the three factors. In an assessment of the hazard of recurrence among the groups, we finally classified the population into 3 classes: class 1 (groups 1), class 2 (groups 2–8), and class 3 (groups 9–12) (Fig 1B). The survival curves of the 12 groups and 3 classes are shown in Fig 1C and 1D, which suggested that the classification effectively differentiated the population according to the potential for tumor recurrence.

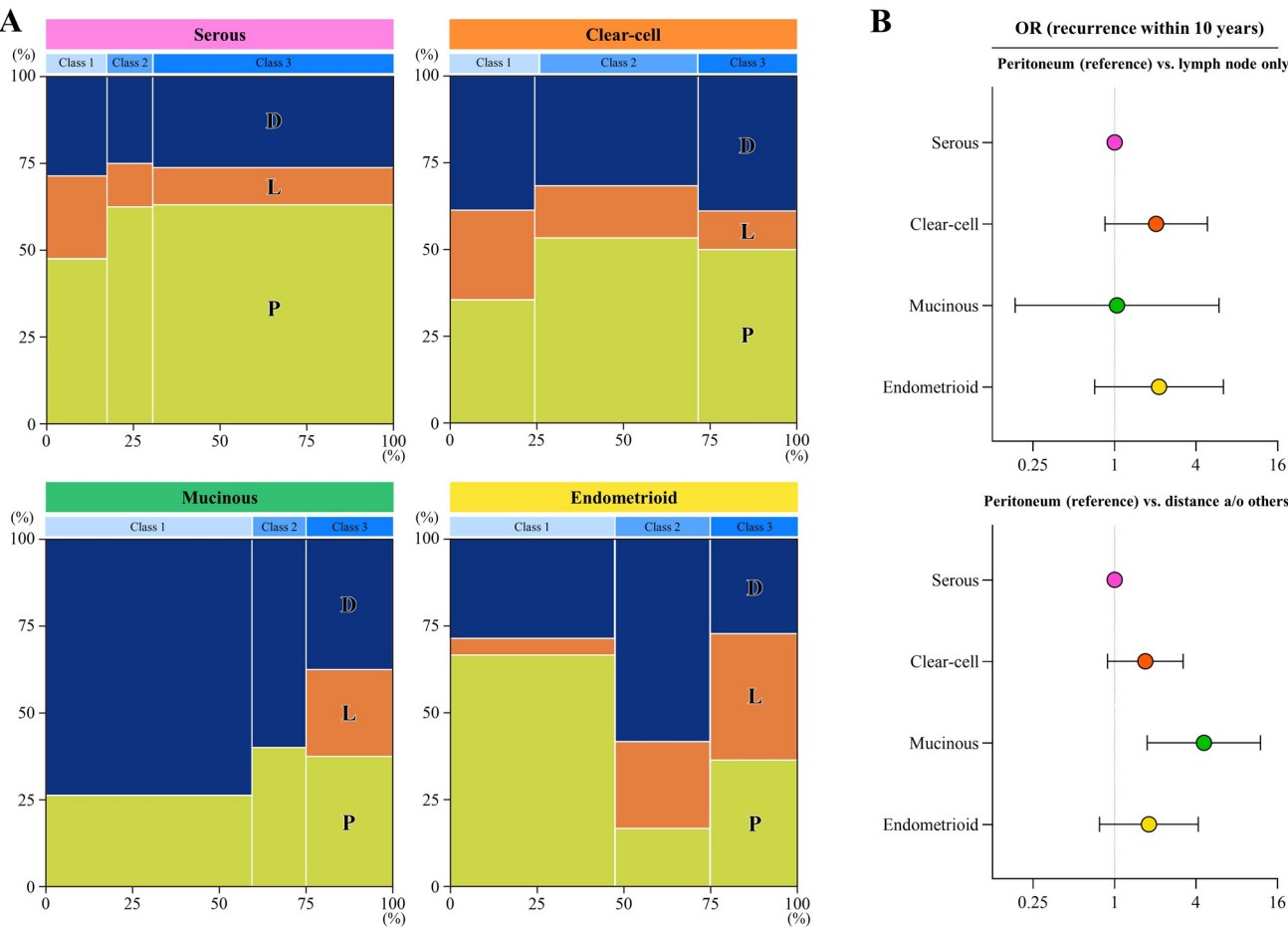

**Fig 3. Effects of histology types on the site of recurrence.** Distributions of the site of recurrence stratified by the 3 classes in patients belonging to each histology type are shown (Marimekko charts) (A). P: Peritoneal metastasis, L: Lymph node metastasis, D: Distant and/or other metastasis. Odds ratios for recurrence within 10 years (peritoneum vs. lymph node; peritoneum vs. distance and/or others) compared among histology types were estimated by a multinominal logistic regression analysis adjusted by age, stage, uterine-preserving surgery, full-staging lymphadenectomy, positive ascites cytology, ascites volume, and chemotherapy (B).

### Effects of histology types for recurrence-free survival

We evaluated the impact of histology types on recurrence-free survival in each class described above. Overall, the survival outcomes varied significantly by class for all four histology types (Fig 2A). The risk of recurrence was higher for the serous histology than for the other histologies (serous: hazard ratio [HR] = 3.406; 95% confidence interval [CI] = 2.412–4.812; P < 0.001, endometrioid: reference). In class 2, the serous and clear-cell histologies showed a slightly worse survival course than the other two (serous: HR = 1.79; 95% CI = 0.98–3.268; P = 0.058, clear: HR = 2.134; 95% CI = 1.274–3.576; P = 0.004, endometrioid: reference, S2A Fig). On the other hand, in mucinous histology, class 3 showed a poor prognosis and affected survival outcomes. Regarding the recurrence-free interval in patients who developed recurrent tumors, no significant differences were observed among the histologies (Figs 2B and S2B). Generally, patients in class 3 developed earlier tumor recurrence than those in the class 1 and the trend was significant in the serous and the endometrioid histologies. Collectively, these results suggested that histology types with a clinical classification provided insights into the risk and timing of recurrence in patients with OvCa who achieved complete tumor resection in the initial surgery.

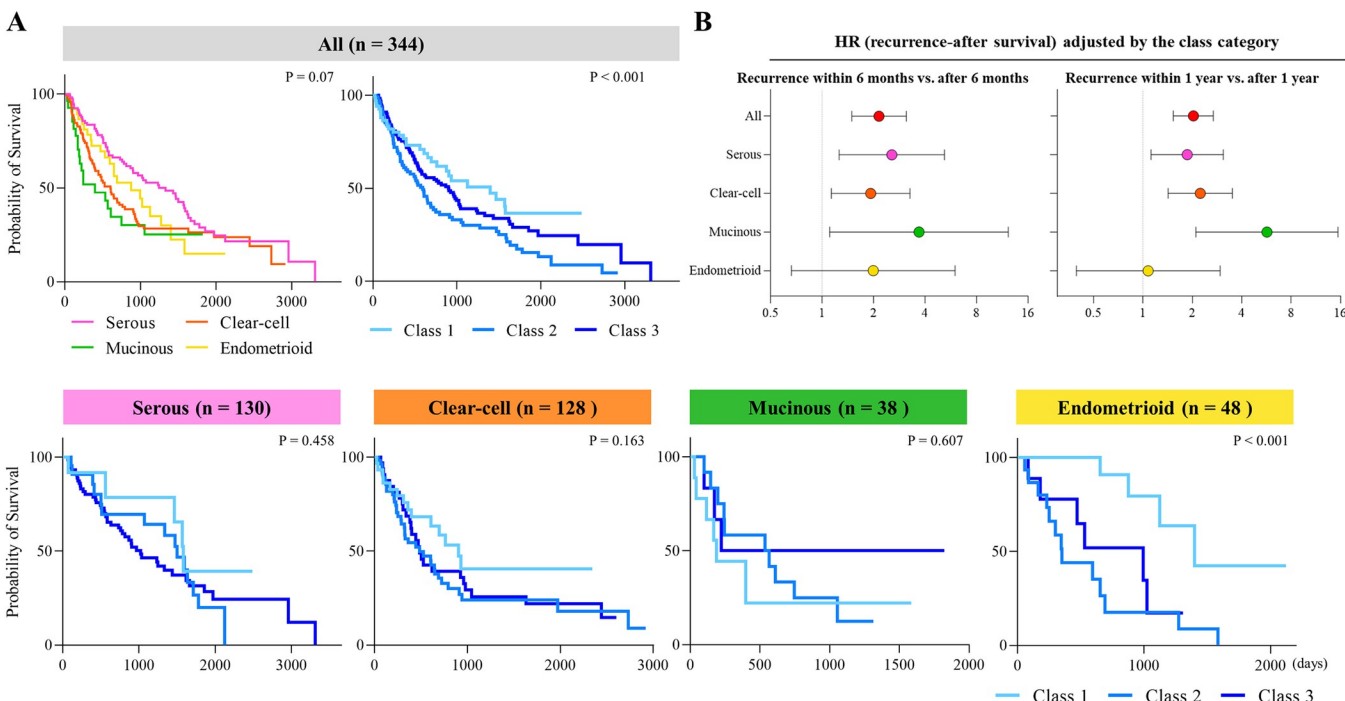

**Fig 4. Effects of histology types on post-recurrence survival.** Kaplan-Meier curves of post-recurrence survival stratified by histology types and the 3 classes (A,B). P-values were estimated by the Log-rank test. Histology-specific hazard ratios of the time to recurrence (within 6 months vs. after 6 months; within 1 year vs. after 1 year) for post-recurrence survival were assessed by a multivariate Cox regression analysis adjusted by the classes (C).

## Effects of histology types on the site of recurrence

We examined the impact of the histology type on the site of recurrence in terms of peritoneal, lymph node, and distant metastasis. Recurrence was detected in serous histology: 45.7% (n = 121), clear-cell: 27.6% (n = 127), mucinous: 18.9% (n = 32), and endometrioid: 15.7% (n = 44). The recurrence cases were divided into three groups: those with peritoneal dissemination (P), those with lymph node metastasis only (L), and those with distant metastasis only (D). There were 164 cases of P overall, of which 16 had lymph node metastasis, 27 had distant metastasis, and 4 had lymph node and distant metastasis (i.e., all site). We visualized differences in the sites of recurrence on a Marimekko chart stratified by histology in Fig 3A. The images obtained indicated that serous and clear-cell histologies were more likely to develop peritoneal metastasis, while mucinous histology preferably caused distant metastasis. Mucinous histology did not present lymph node metastasis in class 1 or 2. In the multinominal logistic regression analysis, mucinous histology was significantly more likely to develop recurrent tumors at a distant site from the peritoneum than other histologies (mucinous: OR = 4.575; 95% CI = 1.746–11.986; P = 0.002, serous: reference, Fig 3B and Table 2). On the other hand, endometrioid histology more frequently developed lymph node metastasis in classes 2 and 3 than other histologies (Fig 3A). Collectively, these results suggested that histology types also affected the site of recurrent OvCa after macroscopically complete tumor resection.

## Effects of histology types on post-recurrence survival

We investigated how histology types affected the prognosis of patients, namely, post-recurrence survival (Fig 4A). The results indicated that the classes affected even prognosis of post-recurrence survival. In contrast to the results obtained on recurrence-free survival, serous

**Table 1. Baseline characteristics of patients in this study cohort.**

| Categories | Epithelial OvCa (n = 1,175) |
|---|---|
| Histology, n (%) | |
| Serous | 265 (22.6) |
| Clear-cell | 461 (39.2) |
| Mucinous | 168 (14.3) |
| Endometrioid | 281 (23.9) |
| Age, years | 54.2 ± 12.0 |
| Stage, n (%) | |
| IA | 264 (22.5) |
| IB | 10 (0.9) |
| IC1 | 294 (25.0) |
| IC2 | 66 (5.6) |
| IC3 | 184 (15.7) |
| IIA | 41 (3.5) |
| IIB | 88 (7.5) |
| IIIA | 32 (2.7) |
| IIIB | 41 (3.5) |
| IIIC | 155 (13.2) |
| CA-125, IU/mL | 703.1 ± 3319.3 |
| Surgical procedure, n (%) | |
| Uterine-preserving surgery | 141 (12.0) |
| Full-staging lymphadenectomy | 636 (54.1) |
| Positive ascites cytology, n (%) | 360 (30.6) |
| Ascites volume, n (%) | |
| <100 mL | 895 (76.2) |
| ≥100 mL | 280 (23.8) |
| Chemotherapy, n (%) | 967 (82.3) |

Data are presented as the mean ± standard deviation or as a proportion (%).

Abbreviations: OvCa, ovarian cancer; CA, cancer antigen.

histology showed a favorable prognosis for post-recurrence survival, especially in classes 1 and 2 (Log-rank P < 0.001, Fig 4). However, patients with recurrent clear-cell and mucinous carcinoma had a poorer prognosis than those with serous histology, irrespective of class. Overall, the classification affected survival outcomes, even post-recurrence survival (Figs 4A and S3). We also examined how the time to recurrence impacts the overall survival in different histology types. We demonstrated that the patients with recurrence within 6 months and 1 year had a poorer prognosis than those with recurrence after 6 months and 1 year. However, this trend was not observed in patients with endometrioid tumors (Fig 4B). In summary, histology types were associated with long-term survival outcomes, not only the time to tumor recurrence, but also post-recurrence survival, in OvCa patients who underwent complete tumor resection at the initial surgery.

## Discussion

The present results demonstrated differences in the behavior of OvCa over a long follow-up period dependent on its histology types. Besides, clinically similar properties common among the histology types were also identified. They appeared to share specific characteristics from other intra-abdominal neoplasms, such as gastric, colorectal, and pancreatic cancers. This is

**Table 2. Multinominal logistic regression analysis for assessing factors associated with site of metastasis (n = 324).**

| Categories | Peritoneum (reference) vs. lymph nodes only | | Peritoneum (reference) vs. distance and/or others | |
|---|---|---|---|---|
| | OR (95%CI) | P value | OR (95%CI) | P value |
| Histology | | | | |
| Serous | reference | | reference | |
| Clear-cell | 2.028 (0.848–4.852) | 0.112 | 1.686 (0.885–3.211) | 0.112 |
| Mucinous | 1.041 (0.184–5.898) | 0.963 | 4.575 (1.746–11.986) | 0.002 |
| Endometrioid | 2.134 (0.714–6.375) | 0.175 | 1.795 (0.774–4.161) | 0.173 |
| Age | | | | |
| <50 years | reference | | reference | |
| ≥50 years | 0.686 (0.321–1.469) | 0.332 | 0.855 (0.490–1.493) | 0.582 |
| Stage | | | | |
| I | reference | | reference | |
| II | 0.444 (0.132–1.494) | 0.190 | 0.777 (0.345–1.754) | 0.544 |
| III | 0.610 (0.262–1.424) | 0.253 | 0.757 (0.406–1.411) | 0.381 |
| CA-125* | 1.254 (1.008–1.561) | 0.042 | 1.060 (0.903–1.246) | 0.475 |
| Surgery | | | | |
| Uterine-preserving surgery | 0.281 (0.069–1.145) | 0.077 | 0.609 (0.246–1.505) | 0.282 |
| Full-staging lymphadenectomy | 0.597 (0.294–1.212) | 0.154 | 0.854 (0.500–1.457) | 0.562 |
| Positive ascites cytology | 1.179 (0.537–2.423) | 0.655 | 0.984 (0.580–1.670) | 0.952 |
| Ascites volume | | | | |
| <100 mL | reference | | reference | |
| ≥100 mL | 0.579 (0.275–1.221) | 0.151 | 0.753 (0.438–1.293) | 0.304 |
| Chemotherapy | 0.434 (0.120–1.570) | 0.203 | 0.799 (0.302–2.115) | 0.652 |

Abbreviations: OR, odd ratio; CA, cancer antigen.

* Logarithmically transformed when analyzed.

the first study to examine histology-specific survival outcomes in OvCa patients without macroscopic residual tumors, which was considered to be clinically significant.

In the first analysis, histology-independent significant prognostic factors were identified in patients without macroscopic residual tumors. Despite the impact of stage, ascites cytology and volume independently affected prognosis. Positive ascites cytology was previously identified as a significant prognostic factor in stage I to III OvCa [15], and its impact was consistent in this cohort. Also, previous studies have shown that ascites volume was associated with the survival outcomes of OvCa [16,17]. Thus, prognosis of patients who achieved macroscopic tumor resection may be effectively classified considering these three factors, and this information will be helpful for clinical practice.

Differences in prognosis by histology, reported previously [18,19], were confirmed in this study. Recurrence-free survival varied for each histology type, even in patients without macroscopic residual tumors. The prognosis was poorest in patients categorized in class 3, likely due to a pre-existing tumor burden. Overall, serous histology showed the highest recurrence rate, while endometrioid histology generally had a more favorable prognosis. Mucinous histology, in particular, negatively impacted the prognosis in class 3. Time to recurrence did not show marked differences. On the other hand, marked differences were observed in the sites of recurrence among the histology types. Mucinous carcinoma in classes 1–2 had less lymph node recurrence and was mostly found in distant metastases without peritoneal dissemination. The multinomial logistic regression analysis showed that the mucinous histology significantly

increased distant metastasis more than peritoneal dissemination. In view of these clinical characteristics, it seems necessary to consider an observation plan dependent on the histology of OvCa.

Regarding post-recurrence survival, patients with each histology type followed different clinical courses. Overall, serous histology had a favorable prognosis, possibly due to the effectiveness of platinum-based chemotherapy. In contrast, mucinous and clear-cell histologies, less respond well to chemotherapy [20,21], showed poorer prognoses. Since the time to recurrence was an important prognostic indicator [22], recurrence within 6 months/1 year was associated with worse outcomes than their counterparts in serous, clear, and mucinous histologies. The endometrioid histology exhibited unique properties showing no differences in prognosis between recurrence within 6 months/1 year and later. Therefore, it is prudent to consider histology types without universally classifying the prognosis of patients based on the time to recurrence.

Complete tumor resection for patients with advanced OvCa is found to improve and prolongs prognosis, suggesting the necessity of maximal tumor reduction in ovarian cancer surgery. [4,23,24]. In patients with intestinal invasion, partial intestinal resection and reconstruction are aggressively recommended [25,26]. Even in patients with hepatobiliary metastasis, conventionally considered as a criterion of unresectability, may require complex multivisceral, but in selected patients, complete cytoreduction has been reported to provide a survival benefit and an acceptable morbidity [27]. Particularly in chemoresistant histologies, complete resection for resectable metastases may be important. Therefore, a careful evaluation of patients and a multidisciplinary approach, considering risks and benefits of complex surgical procedures, are crucial in advanced OvCa patients.

The role of systematic lymphadenectomy in ovarian cancer also warrants discussion. In early-stage OvCa, systematic lymphadenectomy helps detect microscopic nodal metastasis and identify patients who may benefit from adjuvant treatments. However, no definitive evidence suggests a survival benefit of lymphadenectomy in early-stage OvCa [28]. In advanced OvCa, without bulky lymph nodes, systematic lymphadenectomy provides no better outcomes and higher complication and mortality rates [28,29]. Therefore, systematic lymphadenectomy in OvCa is likely to correlate with procedure-related morbidity, but not with survival benefits.

In this study's patient population, lymphadenectomy was performed in over half of the cases (Table 1). Patients were classified by stage, ascites cytology, and ascites volume, but lymph node status was not included. Stage IIIA staging includes positive retroperitoneal lymph nodes (IIIA1) or microscopic dissemination outside the pelvis (IIIA2). We reviewed the data in detail and found that among the 32 stage IIIA cases, 9 had only positive retroperitoneal lymph nodes (IIIA1) and 23 had microscopic dissemination (IIIA2, including one lymph node-positive case). Although small in number and with a short follow-up period, no deaths were observed in IIIA1 cases (9 cases). However, in IIIA2 (23 cases), 10 deaths (43.5%) were observed, similar to those in IIIB and IIIC. Thus, including lymph node status in the classification might have led to more accurate analysis. Nevertheless, since significant differences were confirmed in comparisons of stage I with stage II and stage IIIA, the impact on analyzing histology-specific prognostic and recurrence profile is considered minimal.

Another limitation of this study is that the results are inconclusive because of its retrospective nature and patient accumulation from multiple institutions over a long time. In addition, we were unable to evaluate explicit information on salvage chemotherapy and secondary cytoreductive surgery. However, the strength of this study is the inclusion of a central pathological review system, which led to reduced interobserver variability in histology type identification. Moreover, the initial surgery and treatment were performed based on similar strategies across different institutions. Additionally, the adjuvant chemotherapy regimens were well-defined by original study protocols based on the standard treatment.

In conclusion, long-term oncologic outcomes varied markedly among different histologies in OvCa patients who have undergone complete tumor resection. Therefore, clinicians need to carefully evaluate their clinical backgrounds in order to assess their prognosis. Further clinical research, focused on individualized approaches for OvCa, is essential and will contribute to the development of optimal strategies for managing this lethal gynecologic malignancy.

Availability of data and materials: The data that support the results of this study are available from Nagoya University, but restrictions apply to the availability of these data, which were used under license for the present study, and so are not publicly available.

## Supporting information

**S1 Fig. Flowchart of the selection of patients.** Flowchart of the selection of patients with OvCa who underwent complete tumor resection at the initial surgery from the database of the Tokai Ovarian Tumor Study Group.
(TIF)

**S2 Fig. Related to Fig 1.** Kaplan-Meier curves of recurrence-free survival stratified by histology types and the 3 classes (A). P-values were estimated by the Log-rank test. Recurrence-free intervals for patients who developed recurrent tumors were compared among the histology types with the stratification of the 3 classes (B). P-values were estimated by the Kruskal-Wallis test.
(TIF)

**S3 Fig. Related to Fig 4.** Kaplan-Meier curves of post-recurrence survival stratified by histology types and the 3 classes.
(TIF)

**S1 Table. Cox regression analysis for assessing factors associated with recurrence-free survival (n = 1,175).** Abbreviations: HR, hazard ratio; CA, cancer antigen. * Logarithmically transformed when analyzed.
(DOCX)

## Acknowledgments

We sincerely thank members belonging to TOTSG-affiliated institutions for collaborating in data collection. We sincerely thank Drs. H. Oguchi (TOYOTA Memorial Hospital), K. Sakakibara (Okazaki Municipal Hospital), A. Takeda (Gifu Prefectural Tajimi Hospital), K. Mizuno (Japanese Red Cross Aichi Medical Center Nagoya Daiichi Hospital), O. Yamamuro (Japanese Red Cross Aichi Medical Center Nagoya Daini Hospital), T. Misawa (Nagoya Ekisaikai Hospital), K. Shimizu (Nagoya Ekisaikai Hospital), M. Kawai (Toyohashi Municipal Hospital), K. Umemura (Toyohashi Municipal Hospital), T. Suzuki (Anjo Kosei Hospital), T. Umezu (Kariya Toyota General Hospital), M. Ito (Kasugai Municipal Hospital), S. Morikawa (Komaki City Hospital), R. Onoda (Shizuoka Saiseikai General Hospital), H. Nakamura (Gifu Prefectural Tajimi Hospital), T. Furui (Ogaki Municipal Hospital), T. Sakakibara (Tsushima City Hospital), and M. Hironaka (Nagoya Memorial Hospital) who collaborated in data collection. We sincerely thank Dr. T. Nagasaka who collaborated in the central pathological review.

## Author Contributions

**Conceptualization:** Kazumasa Mogi, Masato Yoshihara.

**Data curation:** Kazumasa Mogi, Masato Yoshihara.

**Formal analysis:** Kazumasa Mogi, Masato Yoshihara, Ryo Emoto, Emiri Miyamoto, Hiroki Fujimoto, Kaname Uno, Sho Tano, Shohei Iyoshi, Kazuhisa Kitami, Nobuhisa Yoshikawa.

**Investigation:** Kazumasa Mogi, Masato Yoshihara.

**Methodology:** Ryo Emoto.

**Project administration:** Masato Yoshihara.

**Software:** Masato Yoshihara, Ryo Emoto.

**Supervision:** Shigeyuki Matsui, Hiroaki Kajiyama.

**Validation:** Kazumasa Mogi, Masato Yoshihara, Ryo Emoto, Emiri Miyamoto, Hiroki Fujimoto, Kaname Uno, Sho Tano, Shohei Iyoshi, Kazuhisa Kitami.

**Visualization:** Masato Yoshihara.

**Writing – original draft:** Kazumasa Mogi, Masato Yoshihara, Emiri Miyamoto, Hiroki Fujimoto, Kaname Uno, Sho Tano, Shohei Iyoshi, Kazuhisa Kitami, Nobuhisa Yoshikawa, Shigeyuki Matsui, Hiroaki Kajiyama.

**Writing – review & editing:** Kazumasa Mogi, Masato Yoshihara, Ryo Emoto, Emiri Miyamoto, Hiroki Fujimoto, Kaname Uno, Sho Tano, Shohei Iyoshi, Kazuhisa Kitami, Nobuhisa Yoshikawa, Shigeyuki Matsui, Hiroaki Kajiyama.

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
