## [Decision Letter · Decision Letter 0]

28 Nov 2023

PONE-D-23-23578Histology-specific long-term oncologic outcomes in patients with epithelial ovarian cancer who underwent complete tumor resection: The implication of occult seeds after initial surgeryPLOS ONE

Dear Dr. Yoshihara,

Thank you for submitting your manuscript to PLOS ONE. After careful consideration, we feel that it has merit but does not fully meet PLOS ONE’s publication criteria as it currently stands. Therefore, we invite you to submit a revised version of the manuscript that addresses the points raised during the review process.

We look forward to receiving your revised manuscript.

Kind regards,

Federico Romano, M.D., Ph.D.

Academic Editor

PLOS ONE

Reviewers' comments:

Reviewer's Responses to Questions

**Comments to the Author**

1. Is the manuscript technically sound, and do the data support the conclusions?

Reviewer #1: Yes

Reviewer #2: Yes

2. Has the statistical analysis been performed appropriately and rigorously? 

Reviewer #1: Yes

Reviewer #2: No

3. Have the authors made all data underlying the findings in their manuscript fully available?

Reviewer #1: Yes

Reviewer #2: Yes

4. Is the manuscript presented in an intelligible fashion and written in standard English?

Reviewer #1: Yes

Reviewer #2: Yes

5. Review Comments to the Author

Reviewer #1: I read with great interest the Manuscript titled " Histology-specific long-term oncologic outcomes in patients with epithelial ovarian cancer who underwent complete tumor resection: The implication of occult seeds after initial surgery”, topic interesting enough to attract readers' attention.

Authors should clarify some point and improve the discussion citing relevant and novel key articles about the topic:

- I suggest a round of language revision, in order to correct few typos and improve readability.

- Considering topic and results if this study, it would be interesting to add few lines about

current evidence of the role of lymphadenectomy in advanced ovarian cancer, considering the advantages and limitations of this procedure. I would be glad if the authors discuss this important point, referring to PMID: 32036457

- In advanced ovarian cancers patients is crucial evaluate morbidity and mortality of cytoreduction considering risk and benefits of this surgical complex procedure. Considering results of this study, the authors should discuss solid evidence about hepatobiliary involvement and the high complex multivisceral surgery it requires. I would be glad if the authors discuss this important point, referring to PMID: 32779050

Considered all this points, I think it could be of interest for the readers and, in my opinion, it deserves the priority to be published after minor revisions

Reviewer #2: 1.The title is histology-specific long-term oncologic outcome of patients with OvCa, but the analyzed factors included stage, ascits, and cytology. To our knowledge,the metastasis of lymph nodes is also included in the staging, and the prognosis is quite different between IIIA or IIIC. So to only difine stage III may not be exact.

2.The results have some predictive value for prognosis, but no clear for treatment.

6. PLOS authors have the option to publish the peer review history of their article (what does this mean?). If published, this will include your full peer review and any attached files.

Reviewer #1: No

Reviewer #2: No

---

## [Author Response · Author response to Decision Letter 0]

26 Jan 2024

Reviewer #1: I read with great interest the Manuscript titled " Histology-specific long-term oncologic outcomes in patients with epithelial ovarian cancer who underwent complete tumor resection: The implication of occult seeds after initial surgery”, topic interesting enough to attract readers' attention.

Authors should clarify some point and improve the discussion citing relevant and novel key articles about the topic:

- I suggest a round of language revision, in order to correct few typos and improve readability.

- Considering topic and results if this study, it would be interesting to add few lines about

current evidence of the role of lymphadenectomy in advanced ovarian cancer, considering the advantages and limitations of this procedure. I would be glad if the authors discuss this important point, referring to PMID: 32036457

- In advanced ovarian cancers patients is crucial evaluate morbidity and mortality of cytoreduction considering risk and benefits of this surgical complex procedure. Considering results of this study, the authors should discuss solid evidence about hepatobiliary involvement and the high complex multivisceral surgery it requires. I would be glad if the authors discuss this important point, referring to PMID: 32779050

Considered all this points, I think it could be of interest for the readers and, in my opinion, it deserves the priority to be published after minor revisions

A: Thank you very much for your valuable feedback. 

We have added the topics and references for lymphadenectomy and cytoreduction in advanced ovarian cancers patients to the discussion as you suggested. And we have corrected typos and omissions to improve readability. In addition, some colors in the graphs of Figures S2 and S3 have been corrected.

Reviewer #2: 1. The title is histology-specific long-term oncologic outcome of patients with OvCa, but the analyzed factors included stage, ascits, and cytology. To our knowledge,the metastasis of lymph nodes is also included in the staging, and the prognosis is quite different between IIIA or IIIC. So to only difine stage III may not be exact.

A: Thank you for pointing this out. As you pointed out, the prognosis may differ between stage IIIA and stage IIIC. We analyzed our data and found that the prognosis tended to be worse in stage IIIC, although the difference was not significant (p = 0.9345).

We reviewed the data in detail and found that there were 32 stage IIIA cases, of which 9 had only positive lymph nodes (IIIA1) and 23 had microscopic dissemination (IIIA2, including one lymph node-positive case).

IIIA1 (9 cases) were small in number and had a short follow-up period, but no deaths were observed. On the other hand, in IIIA2 (23 cases), 10 deaths (43.5%) were observed, similar to those in IIIB and IIIC.

On the other hand, significant differences are confirmed in the comparison between Stage I, II and IIIA (p < 0.001).

Therefore, although positive lymph node metastasis may affect prognosis, it is assumed that peritoneal dissemination may be a worse prognostic factor. 

In this study, we categorized patients using ascites volume and ascites cytology as clinical factors that are potentially related to the dissemination progression.

Although it may be more accurate to add information on lymph node metastasis and dissemination, since this study was conducted to examine the prognostic and recurrence profile of each histology in patients who had complete resection, we think that the effect of stage IIIA1 on those analyses is small.

However, stage IIIA1 cases may have a unique course, and we would like to consider this issue in cases with only positive lymph nodes metastasis as a topic for future research. Thank you very much for your valuable comments. We have added the above information to Limitation.

2.The results have some predictive value for prognosis, but no clear for treatment.

A: Thank you for your precious comments. We think that this study is a retrospective analysis and is not suitable for prospective treatment prediction. From the real-world data in this study, we found ascites volume, ascites cytology, and stage may be useful in predicting prognosis. We also think that our data revealed the recurrence patterns and prognostic characteristics of post-recurrence treatment for each histology. We hope that these data will be helpful in clinical practice.

---

## [Decision Letter · Decision Letter 1]

29 Apr 2024

PONE-D-23-23578R1Histology-specific long-term oncologic outcomes in patients with epithelial ovarian cancer who underwent complete tumor resection: The implication of occult seeds after initial surgeryPLOS ONE

Dear Dr. Yoshihara,

Thank you for submitting your manuscript to PLOS ONE. After careful consideration, we feel that it has merit but does not fully meet PLOS ONE’s publication criteria as it currently stands. Therefore, we invite you to submit a revised version of the manuscript that addresses the points raised during the review process.

We look forward to receiving your revised manuscript.

Kind regards,

Federico Ferrari, MD, PhD

Academic Editor

PLOS ONE

Journal Requirements:

Additional Editor Comments:

Please find the cooments of the Reviewer 2 and try to amend or justify as appropriate.

Reviewers' comments:

Reviewer's Responses to Questions

**Comments to the Author**

1. If the authors have adequately addressed your comments raised in a previous round of review and you feel that this manuscript is now acceptable for publication, you may indicate that here to bypass the “Comments to the Author” section, enter your conflict of interest statement in the “Confidential to Editor” section, and submit your "Accept" recommendation.

Reviewer #1: All comments have been addressed

Reviewer #2: (No Response)

2. Is the manuscript technically sound, and do the data support the conclusions?

Reviewer #1: Yes

Reviewer #2: Partly

3. Has the statistical analysis been performed appropriately and rigorously? 

Reviewer #1: Yes

Reviewer #2: Yes

4. Have the authors made all data underlying the findings in their manuscript fully available?

Reviewer #1: Yes

Reviewer #2: Yes

5. Is the manuscript presented in an intelligible fashion and written in standard English?

Reviewer #1: Yes

Reviewer #2: Yes

6. Review Comments to the Author

Reviewer #1: The quality of the manuscript has improved thanks to the changes made. I think it could be of interest to the readers and, in my opinion, it deserves the priority to be published.

Reviewer #2: The report has some predictive value for prognosis of epithelial ovarian cancer. Except for the lymphadnectomy, the author also raised up that chemotherapy has no relationship with the recurrence of the disease. I think we need more informations or subgroup analysis to explain the result.

7. PLOS authors have the option to publish the peer review history of their article (what does this mean?). If published, this will include your full peer review and any attached files.

Reviewer #1: No

Reviewer #2: No

---

## [Author Response · Author response to Decision Letter 1]

2 May 2024

Additional Editor Comments:

Please find the cooments of the Reviewer 2 and try to amend or justify as appropriate.

A: Thank you for your suggestion, we have responded to reviewer 2's comments.

Reviewer #1: The quality of the manuscript has improved thanks to the changes made. I think it could be of interest to the readers and, in my opinion, it deserves the priority to be published.

A: Thank you for your comment. 

It has improved thanks to your appreciated suggestions.

Reviewer #2: The report has some predictive value for prognosis of epithelial ovarian cancer. Except for the lymphadnectomy, the author also raised up that chemotherapy has no relationship with the recurrence of the disease. I think we need more informations or subgroup analysis to explain the result.

A: Thank you for your valuable comments. In the multivariate analysis of this cohort, chemotherapy was not among the factors that significantly influenced recurrence-free survival. However, this does not mean that chemotherapy has no effect on recurrence. Because this cohort included stage I, II and III patients with no residual tumor, we identified three factors that we thought were common to these patients and used them to stratify the cohort. Therefore, it is difficult to analyze the relationship between chemotherapy, relapse, and histologic type in this cohort and subgroup. 

However, the impact of chemotherapy on microscopic occult tumor metastasis is an important concern, and we would like to investigate the proposed issue in a further study.

---

## [Editor Report · Decision Letter 2]

24 May 2024

PONE-D-23-23578R2Histology-specific long-term oncologic outcomes in patients with epithelial ovarian cancer who underwent complete tumor resection: The implication of occult seeds after initial surgeryPLOS ONE

Dear Dr. Yoshihara,

Thank you for submitting your manuscript to PLOS ONE. After careful consideration, we feel that it has merit but does not fully meet PLOS ONE’s publication criteria as it currently stands. Therefore, we invite you to submit a revised version of the manuscript that addresses the points raised during the review process.

We look forward to receiving your revised manuscript.

Kind regards,

Federico Ferrari, MD, PhD

Academic Editor

PLOS ONE
---

## [Author Response · Author response to Decision Letter 2]

30 May 2024

Thank you very much for your suggestion.

We have reviewed the reference list and verified that it is complete and correct.

We uploaded Figures 1-4 to PACE and no image problem was detected.

Thanks.

---

## [Editor Report · Decision Letter 3]

8 Sep 2024

PONE-D-23-23578R3Histology-specific long-term oncologic outcomes in patients with epithelial ovarian cancer who underwent complete tumor resection: The implication of occult seeds after initial surgeryPLOS ONE

Dear Dr. Yoshihara,

Thank you for submitting your manuscript to PLOS ONE. After careful consideration, we feel that it has merit but does not fully meet PLOS ONE’s publication criteria as it currently stands. Therefore, we invite you to submit a revised version of the manuscript that addresses the points raised during the review process.

We look forward to receiving your revised manuscript.

Kind regards,

Alison May Berner

Academic Editor

PLOS ONE

Journal Requirements:

Additional Editor Comments:

Please perform a final proof read of the document and correct typographic errors.

Examples where these currently occur include:

- Abstract

- Line 267

- Title of Reference List

---

## [Author Response · Author response to Decision Letter 3]

16 Sep 2024

Thank you very much for your pointing.

We have corrected typographical errors in the manuscript.

We have also removed duplicate references and renumbered references in the reference list.

We uploaded Figures 1-4 to PACE and no image problem was detected.

Thanks.

---

## [Editor Report · Decision Letter 4]

18 Sep 2024

Histology-specific long-term oncologic outcomes in patients with epithelial ovarian cancer who underwent complete tumor resection: The implication of occult seeds after initial surgery

PONE-D-23-23578R4

Dear Dr. Yoshihara

We’re pleased to inform you that your manuscript has been judged scientifically suitable for publication and will be formally accepted for publication once it meets all outstanding technical requirements.

Kind regards,

Alison May Berner

Academic Editor

PLOS ONE

Additional Editor Comments (optional):

There remain some small typos (e.g. the "References" title") which need to resolved at the proof stage.

Reviewers' comments:

None additional

---

## [Editor Report · Acceptance letter]

23 Sep 2024

PONE-D-23-23578R4 

PLOS ONE

Dear Dr. Yoshihara, 

I'm pleased to inform you that your manuscript has been deemed suitable for publication in PLOS ONE. Congratulations! Your manuscript is now being handed over to our production team.

Kind regards, 

on behalf of

Dr. Alison May Berner 

Academic Editor

PLOS ONE